TGF-β-mediated activation of fibroblasts in cervical cancer: implications for tumor microenvironment and prognosis

Qu Haina
Zhao Jing
Zuo Xia
He Hongyue
Wang Xiaohan
Li Huiyan
Zhang Kun zhangkun328@126.com
Obstetrics and Gynecology Department, Xi’an People’s Hospital (Xi’an Fourth Hospital) , Xi’an , China
Wang Jincheng
Electronic publication date: 2025 Mar 19
Publication date: 2025
Volume: 13
Electronic Location ID: e19072
Received 2024 Oct 29; Accepted 2025 Feb 10
Copyright: ©2025 Qu et al.
Copyright year: 2025
Copyright holder: Qu et al.
License: This is an open access article distributed under the terms of the Creative Commons Attribution License, which permits unrestricted use, distribution, reproduction and adaptation in any medium and for any purpose provided that it is properly attributed. For attribution, the original author(s), title, publication source (PeerJ) and either DOI or URL of the article must be cited.
License URL: https://creativecommons.org/licenses/by/4.0/

Keywords: TGF-β  signaling pathway, Cervical cancer, Fibroblasts, Immunity, Prognosis

Funding: The authors received no funding for this work.

==============================
Background

Cervical cancer (CC) is a prevalent female malignancy strongly influenced by the tumor microenvironment (TME). This study focuses on the role of TGF-β signaling in cancer-associated fibroblasts (CAFs) and its interaction with immune cells, aiming to elucidate its impact on CC progression.

Methods

The TME of CC patients was analyzed using scRNA-seq data and we identified the major cell types in the TME with a focus on the activation of the TGF-β signaling pathway in fibroblasts. Gene modules related to the TGF-β signaling pathway were identified by Weighted correlation network analysis (WGCNA). Using The Cancer Genome Atlas Cervical Squamous Cell Carcinoma and Endocervical Adenocarcinoma (TCGA-CESC) dataset, a prognostic gene model was constructed by univariate Cox, LASSO Cox and multivariate Cox regression analyses. For cellular validation, the mRNA level of prognostic model-related genes was tested via quantitative real-time real-time polymerase chain reaction (PCR). Thereafter, the following assays, including cell counting kit-8, scratch and wound healing assays, were applied to assess the viability, migration and invasion of CC cells.

Results

Analysis at single-cell resolution identified nine major cell types in the TME, and significant activation of the TGF-β signaling pathway in fibroblasts was correlated with tumor proliferation and differentiation. Strong TGF-β signaling communication between fibroblasts and macrophages and NK/T cells suggested a crucial role in the shaping of the immunosuppressive microenvironment. WGCNA analysis identified gene modules significantly associated with the TGF-β signaling pathway. The prognostic model constructed based on three genes, ITGA5, SHF and SNRPN, demonstrated good predictive ability in multiple datasets, validating its potential for clinical application. Meanwhile, the cellular validation assays have revealed the higher expression of ITGA5 and SNRPN and lower expression of SHF in CC cells. Further, ITGA5 knockdown suppressed the viability, migration and invasion of CC cells.

Conclusion

This study confirmed the important role of the TGF-β signaling pathway in CC, especially in fibroblasts on tumor microenvironment and tumor progression. The current model could effectively evaluate the prognosis of CC, providing a theoretical foundation for developing CC therapies according to the TGF-β signaling pathway. The present results provide new perspectives for further research on the pathological mechanisms and clinical management of CC.

Introduction

Cervical cancer (CC) is a common female malignancy worldwide (Nisha et al., 2023; Zeng et al., 2024), especially in developing regions. The World Health Organization (WHO) showed that the incidence of new CC cases in 2020 was 604,000 worldwide, causes 342,000 deaths (Cohen et al., 2019; Xu et al., 2022). CC incidence and mortality are more prevalent in less developed nations, accounting for more than 90% of the global CC burden. Surgery, radiotherapy, chemotherapy and immunotherapy are the main treatments for CC. CC patients at an early-stage are usually treated with surgical procedures, such as hysterectomy or conization. For locally advanced CC, radiotherapy combined with chemotherapy is the standard treatment option (Chargari et al., 2022). In recent years, with the development of molecularly targeted drugs and immune checkpoint inhibitors (PD-1/PD-L1 inhibitors), the field of CC treatment has also made significant progress (Mayadev et al., 2022; Ferrall et al., 2021). However, despite major advances in treatment strategies, the prognostic outcomes of patients with advanced or recurrent CC remains unfavorable. Chemotherapy has limited effectiveness and is often accompanied by severe side effects (Yadav, Srinivasan & Jain, 2024). Immunotherapy, despite showing efficacy in some patients, still has a low overall response rate, which is largely due to the complexity of the TME and the presence of immunosuppressive mechanisms (Yadav, Yadav & Alam, 2023).

The TGF-β signaling pathway is closely involved in a variety of cancers, including CC. This signaling pathway normally functions in normal cells to inhibit cell proliferation, promote cell differentiation, and maintain tissue homeostasis (Zhong et al., 2023). However, in the tumor environment, the TGF-β signaling pathway tends to exhibit dual roles, both inhibiting early tumor development and promoting tumor invasion and metastasis in late stages (Fan et al., 2023). In CC, aberrant activation of the TGF-β signaling pathway is closely related to cancer progression. TGF-β initiates downstream SMAD-dependent and non-SMAD-dependent signaling pathways through the activation of its receptors, TGFBR1 and TGFBR2, which regulate cell apoptosis, migration, invasion, and proliferation (Wu et al., 2016). A previous report showed that activation of the TGF-β signaling pathway in CC cells and CAFs promotes epithelial-mesenchymal transition (EMT) and enhances invasiveness and metastasis of tumor cells (Liu et al., 2023). The TGF-β signaling pathway not only affects the behavior of CC cells, but also functions critically in shaping the tumor microenvironment (TME). TGF-β inhibits anti-tumor immune responses and promotes tumor immune escape through interactions with immune cells. For example, TGF-β inhibits the activity of NK cells and CD8+ T cells and enhances the differentiation of immunosuppressive Treg cells, thereby creating an immunosuppressive microenvironment that supports tumor growth (Li et al., 2024; Chen et al., 2021b). Additionally, the TGF-β signaling pathway regulates the activity of CAFs, which play an important role in ECM remodeling and tumor angiogenesis. It has been shown that fibroblasts in cc promote tumor invasion and metastasis by secreting large amounts of TGF-β and activating TGF-β signaling in their own and neighboring cells. This signaling cascade effect makes TGF-β a potential therapeutic target (Chen et al., 2021a). Considering the dual role of the TGF-β signaling pathway in CC, therapeutic strategies targeting this pathway are complicated. Current studies focused on blocking the activity of the TGF-β signaling pathway to inhibit its tumor-promoting effects in advanced tumors. For example, TGF-β receptor inhibitors (Galunisertib) and TGF-β-neutralizing antibodies have shown potential in preclinical studies and early clinical trials (Ghanaatgar-Kasbi et al., 2022). However, given the important role of TGF-β in normal tissue homeostasis, complete inhibition of this signaling pathway may lead to serious side effects. Therefore, future studies may require more precise targeting strategies, such as selective inhibition of TGF-β signaling in tumor cells or specific types of cells in the TME, without interfering with physiological functions in normal tissues. Although significant progress has been made in understanding the role of the TGF-β signaling pathway in CC, there are still many questions that need to be further explored.

This study revealed the aberrant activation of the TGF-β signaling pathway in fibroblasts and its important role in tumor progression by analyzing the multi-omics data of CC patients. Landscape analysis with single-cell resolution identified nine major cell types in the TME, especially the significant role of fibroblasts in CC. This study further identified gene modules significantly related to the TGF-β signaling pathway, which may be important molecular mechanisms driving the malignant phenotype of CC.

Materials and Methods

Data collection and preprocessing

From The Cancer Genome Atlas (TCGA, https://portal.gdc.cancer.gov/) database, cervical squamous cell carcinoma and endocervical adenocarcinoma (CESC) data were downloaded. Samples without survival time or status were removed and all patients were guaranteed to have a survival time greater than 0 days. RNA-seq expression profiles were downloaded and converted to TPM format and log2 transformed. A sum of 291 tumor samples were ultimately used for analysis. The TCGA-CESC dataset is divided into a training set (70%) and a validation set (30%) by means of random partitioning to ensure balanced and scientific model construction and validation.

GSE44001 microarray datasets with survival time were selected and probes from the Gene Expression Omnibus (GEO, https://www.ncbi.nlm.nih.gov/geo/) were converted to Symbol according to the annotation file (Song et al., 2023). Samples without clinical follow-up data and OS data were removed, a total of 300 tumor samples in GSE44001 retained. The single cell data GSE208653 was also downloaded from the GEO website. It contained two normal samples and 3 HPV-infected CC samples. TGF-β signaling-related genes (HALLMARK_TGF_BETA_SIGNALING.v2023.2.Hs.gmt) were obtained from MsigDB (https://www.gsea-msigdb.org/gsea/msigdb).

Single-cell RNA-seq data preprocessing

The Read10X function of the Seurat package (Song et al., 2023; Stuart et al., 2019; Zulibiya et al., 2023) was utilized to read the downstream data, retaining 10% of the mitochondrial genes and cells with gene numbers between 200 and 8,000. The SCTransform function was used for normalization. After principal component analysis (PCA) downscaling, we used the harmony package (Korsunsky et al., 2019) to remove the batch effect between samples (max.iter.harmony=50, lambda=0.5). Next, based on the first 30 principal components, UMAP was performed to reduce the dimensionality, and finally clustered the cells into groups using the FindNeighbors and FindClusters functions. For all cells, resolution=0.1.

Correlation analysis of TGF-β signaling activity with inflammatory pathways, proliferation and metabolism

The HALLMARK_TGF_BETA_SIGNALING.gene collection was downloaded from the MsigDB database. The expression matrix of fibroblasts was extracted. We used the AUCell package to sequentially calculate the AUCell score (Aibar et al., 2017) for the gene collections within each sample. In addition, we extracted the inflammatory, proliferative and metabolic signaling pathways from MsigDB and used the AUCell score to calculate the correlation with the enrichment score of the TGF-β signaling pathway by the Pearson method.

Pseudo-time trajectory analysis

Pseudo-time trajectory analysis of fibroblasts was performed using Monocle. We constructed sets of differentially expressed genes (DEGs) between normal and tumor groups using the FindMarkers function, which was used to construct differentiation trajectories. We used the branch with number of normal cells as the starting end of the trajectory (Du et al., 2023).

Cell communication analysis

The CellChat package (Jin et al., 2021) was used to construct the cellular subpopulation ligand–receptor interaction network, and the netVisual_bubble package was used to show the bubble diagrams of receptors and ligands in different groups of cells and between cells and the number of communications between them. The TGF-β signaling pathway was used to construct its communication in the individual cells as well as the communication of the individual receptor–ligand pairs that were included.

Weighted correlation network analysis

To filter genes related to TGF-β signaling, we performed weighted correlation network analysis (WGCNA) (Langfelder & Horvath, 2008; Song et al., 2023). The WGCNA package was utilized to identify TGF-related gene modules. The samples were clustered to screen for co-expression modules, and to ensure a scale-free network, the appropriate soft threshold β was selected by the pickSoftThreshold function. Gene module identification was performed by hierarchical clustering and similar gene modules were merged with at least 60 genes per module (minModuleSize = 60). Then the correlation between modules and feature scores was assessed. Finally the genes in the modules with higher correlation were obtained for subsequent analysis.

CC prognostic modeling

The prognostic relevance of key modular genes obtained based on WGCNA was determined by univariate COX regression analysis, the number of genes was narrowed down by LASSO COX regression analysis based on the glmnet package, the key genes and correlation coefficients were obtained through multifactoriality and the risk score was calculated for each patient (Simon et al., 2011). The risk score in the prognostic model can be expressed as: risk score = Σβi × Expi, where β represents the coefficients from the Lasso method and Exp denotes the expression levels of the prognostic genes (i). Then based on the optimal cutoff value of the risk score, low-risk and high-risk groups of patients were classified. The Kaplan–Meier (K-M) survival analysis demonstrated the survival duration between two distinct risk groups. To assess the effectiveness of the prognostic model in forecasting varying survival times, receiver operating characteristic (ROC) curve analysis was conducted using the timeROC R package (Li et al., 2023).

Tumor microenvironment immune cell infiltration analysis

Gene sets of 28 types of immune cells from Charoentong et al. (2017) were extracted and ssGSEA was conducted using GSVA package to calculate the level of immune cell infiltration in the TME of samples in TCGA-CESC (Hanzelmann, Castelo & Guinney, 2013; Charoentong et al., 2017). We also performed to determine the stromal parity and immunity parity in the TME of the CC samples by the ESTIMATE method (Yoshihara et al., 2013). Finally, we calculated the distribution of the six immune cell scores in the TCGA dataset across different risk groups based on the TIMER algorithm (Li et al., 2020).

Cell culture and transfection

Keratinocyte-serum free medium (17005-042; Gibco, Waltham, MA, USA) supplemented with 0.1 ng/mL recombinant epidermal growth factor (P5552; Beyotime, Shanghai, China), 0.05 mg/mL bovine pituitary extract (13028-014; Gibco, USA) and 0.4 mM calcium chloride (ST365; Beyotime) was used to culture human cervix epithelial cell line Ect1/E6E7 (CRL-2614) purchased from American Type Culture Collection (Manassas, MD, USA). Meanwhile, the CC cell line Hela (BNCC342189; BeiNa Culture Bio, Xinyang, China) was cultured in high glucose Dulbecco’s modified Eagle’s medium (11965-092; Gibco) with the supplementation of 10% fetal bovine serum (C0039; Beyotime). All the cells were tested via short tandem repeat profiling and incubated in the incubator with 5% CO2 at 37 °C.

For the liposome transfection, the small interfering RNA against ITGA5 and the control small interfering RNA (siRNA) were all purchased from GenePharma (Shanghai, China) and transfected into Hela cells with lipofectamine 2000 transfection reagent (11668-027; Invitrogen, Carlsbad, CA, USA) as per the manuals. The sequences applied for the transfection were listed in Table 1. In this case, si-NC served as a negative control group, where cells were transfected with non-specific siRNA that did not target specific genes. This group is used to exclude potential effects of the transfection process or non-specific siRNA on cell behavior. si-ITGA5, on the other hand, means that the cells were transfected with siRNA targeting the ITGA5 gene, and this was used to specifically knock down the expression of the ITGA5 gene in order to study the function of ITGA5 in cell proliferation, migration, and invasion.

Table 1 Sequences for the transfection.

Target	Sequences (5′–3′)	
siITGA5	CGGATTCTCAGTGGAGTTTTACC	
siNC	TCACGTTCTCGAGTGGAGTTTAC	

Cell viability assay

Transfected Hela cells were cultured in a 96-well plate at the density of 2 × 103 cells/well for 48-hour culture and treated with 10 µL CCK-8 solution (C0037; Beyotime) for 4-hour culture. The optical density at 450 nm was read in iMark microplate reader (Bio-Rad, Hercules, CA, USA) to calculate the viability of transfected Hela cells (Feng & Xiao, 2024).

EdU test

Hela cells were seeded into 24-well plates. Following the protocol provided with the EdU kit (BeyoClick™ EdU Cell Proliferation Kit with Alexa Fluor 488; Beyotime), EdU solution was added to each well, and the cells were incubated for 4 h before being washed with PBS. The cells were subsequently fixed using 4% paraformaldehyde. Next, Apollo solution (RiboBio) was introduced to incubate the cells in darkness for 30 min, and 0.5% Triton X-100 was included to enhance cell permeability. Finally, the cells were treated with Hoechst 33,342 (1:10,000; Sigma-Aldrich) and examined using a fluorescence microscope.

Cell migration assay

Transfected Hela cells were grown in a 6-well plate with serum-free media and received an artificial scratch on the monolayer using a 200-µL sterile pipette tip once they reached complete confluence. After 48 h, cells were photographed under an inverted optical microscope (DP27; Olympus, Tokyo, Japan) and the wound closure (%) was accordingly quantified to determine the migration of CC cells.

Cell invasion assay

A 24-well transwell plate with polycarbonate membrane (pore: eight µm, 3422; Corning, Inc, Corning, NY, USA) coated with matrix gel (C0372; Beyotime) was used in cell invasion assay. Transfected CC cells were cultured in the upper chamber containing 200 µL serum-free media, while the lower chamber was added with 700 µL culture media containing 10% serum. After 48 h, paraformaldehyde (P0099; Beyotime) was used for fixing the cells, followed by dyeing with 0.1% crystal violet (C0121; Beyotime) for 30 min. Three random fields were observed with an inverted optical microscope (DP27; Olympus, Japan) and the number of invaded cells was quantified.

Quantitative real-time PCR

The TriZol total RNA extraction kit (15596-026; Invitrogen, Waltham, MA, USA) was applied to isolate the total RNA from Hela and Ect1/E6E7 cells according to the manuals. Next, the concentration of isolated RNA was tested, followed by the synthesis of complementary DNA with a relevant assay (D7178S; Beyotime) for the reverse transcription. SYBR Green qPCR Mix (D7260; Beyotime) was thereafter applied for the PCR assay according to the protocols. The relative expression was calculated via the 2−ΔΔCT method with GAPDH as a normalizer (Livak & Schmittgen, 2001). See Table 2 for the sequences of primers applied in this study (Zhang et al., 2023).

Table 2 Sequences of the primers.

Target	Sequences (5′–3′)	
ITGA5 forward	ATTCTCAGTGGAGTTTTACC	
ITGA5 reverse	ATTAAGGATGGTGACATAGC	
SHF forward	TGTATGACACACCCTATGAG	
SHF reverse	GTATGACAGTTGAGGGAGAG	
SNRPN forward	GTGATTGTGATGAGTTCAGA	
SNRPN reverse	ACAGTCATGGATACCAAGTT	
GAPDH forward	ATTGACCTCAACTACATGGT	
GAPDH reverse	CATACTTCTCATGGTTCACA	

Statistical analysis

All the statistical data were analyzed in R language (version 3.6.0; R Core Team, 2019) and GraphPad Prism software (version 8.0.2; GraphPad, Inc., La Jolla, CA, USA). The Wilcoxon rank-sum test and student’s t-test were used to calculate the differences between two-group continuous variables. Correlations were calculated using spearman method. The variability in survival time between each group of patients was compared by log-rank test, and p < 0.05 was defined as statistically different.

Results

Single-cell resolution landscape

There were nine cell types determined in five samples via analysis of single-cell data from the GSE208653 dataset, namely, NK/T cells, neutrophil cells, macrophage cells, plasma cells, endothelial cells, epithelial cells, B cells, mast cells, fibroblast cells (Fig. 1A). The expressions of the marker genes in the nine cell types were demonstrated, where COL1A2, DCN, and COL1A1 showed a remarkable high-level expression trend in fibroblast cells (Figs. 1B–1C).

Figure 1 Single-cell resolution landscape and expressions of marker genes.

(A) Distribution of nine cell types in the sample. (B) Bubble plots demonstrating the expression levels of marker genes in the nine cell types. (C) Violin plot demonstrating the expressions of marker genes in the nine cell types.

The abnormal activation of the TGF-β signaling pathway in cervical cancer fibroblast cells would promote proliferation

In GSE208653, fibroblast cells from normal and CC samples were extracted. TGF-β signaling activity in fibroblast cells of CC samples was dramatically increased over normal samples (Fig. 2A). TGF-β signaling pathway-related genes, ID2, PPP1R15A, SMAD7, and XIAP were significantly hyper-expressed in fibroblast cells from CC samples (Fig. 2B). Pseudo-time analysis was executed on fibroblast cells, and the dark blue segment was identified as the differentiation starting point of fibroblast cells (Fig. 2C). Color coding by sample type showed that fibroblast cells in normal samples were located at the differentiation starting point, and fibroblast cells in CC samples were located at the end of differentiation (Figs. 2D–2E). The expression levels of THBS1, SKL, SLC20A1, KLF10, SMAD7, PPP1R15A, ID2, JUNB, and SERPINE1 followed the pseudo-timeline, with a small incremental increase, followed by a decrease, and eventually a gradual increase, which reached the highest at the terminal end of the pseudo-timeline (Fig. 2F). The expression of the signature markers of fibroblast proliferation, CD74, FN1, were elevated at the end of the pseudo-timeline, meaning that it was elevated in fibroblasts in the CC group (Fig. 2G). We also observed that the cell proliferation signature genes CXCL10, CXCL9, STAT1, and the cell cycle signature gene IRF1 showed the same trend of elevated expression in fibroblasts in the CC group (Figs. 2H–2I). The apoptosis inhibition-related genes HSPA1A and HSPA1B showed a trend of high expression in the high and low TGF-β signaling score groups (Fig. 2J). Cell proliferation-related genes JUN, JUND, and JUNB showed high expressions in the high TGF-β signaling score subgroup (Fig. 2K). We speculated that activating the TGF-β signaling pathway in fibroblasts in the tumor group promoted fibroblast proliferation.

Figure 2 Aberrantly activated TGF BETA signaling pathway in cervical cancer fibroblasts promotes proliferation.

(A) Fibroblast TGF BETA SIGNALING AUCell score in normal and tumor groups. (B) Expression of TGF BETA SIGNALING-related genes in fibroblasts in normal and tumor groups. (C–D) Proposed time-series analysis of fibroblasts. (E) Cell density distribution of fibroblasts with pseudo-timeline in normal and CC groups. (F) The expression heatmap of TGF BETA SIGNALING related genes on the pseudo-time trajectory. (G) The expression heatmap of fibroblast proliferation characterized genes with pseudo-timeline trajectories. (H) Heatmap of cell proliferation characterized genes expression level with pseudo-timeline trajectory. (I) Expression heatmap of cell cycle characterized genes with pseudo-timeline trajectories. (J) Violin plot of expressions of apoptosis inhibition related genes in high and low TGF BETA SIGNALING score groups. (K) Violin plot of expressions of cell proliferation-related genes in the two subgroups.

TGF-β signaling pathway suppresses inflammatory, promotes escape, and regulates metabolism

In CC samples, we assessed the correlation of TGF-β signaling pathway activity with the activity of inflammatory pathways, metabolic pathways, and proliferative pathways. The results indicate that TGF-β signaling pathway activity is significantly and positively correlated with the activity of some inflammatory pathways (e.g., negative regulation of B cells differentiation and immune system processes, negative regulation of CD4 positive alpha beta T cell differentiation, and negative regulation of alpha beta T cell differentiation) (Fig. 3A). The TGF-β signaling pathway showed the same significant positive correlation trend with the proliferation pathway activity and metabolism-related pathways, the proliferation-related pathways included TGF-β signaling, KRAS signaling up, P53 pathway, MTORC1 signaling, and G2M checkpoint, mitotic spindle (Fig. 3B), and the metabolic-related pathways include regulation of phosphorus metabolic process, positive regulation of phosphorus metabolic process, and positive regulation of protein metabolic process (Fig. 3C).

Strong TGF-β signaling communication between fibroblasts and Macrophage cells, NK/T cells

We analyzed the interactions between fibroblasts and seven cell types in the CC group. The results showed that there were strong interactions between fibroblasts and NK/T cells, neutrophil cells, macrophage cells, epithelial cells, mast cells, fibroblast cells, B cells, endothelial cells, especially NK/T cells, but fibroblasts and plasma cells had weak interactions (Fig. 4A). Further analysis of TGF-β signaling intensity between fibroblasts and the seven cell types visualized the phenomenon of strong TGF-β signaling communication between fibroblasts and macrophage cells as well as NK/T cells (Fig. 4B). Evidently, fibroblasts and NK/T cells cells interacted mainly through TGFB3-(TGFBR1+TGFBR2), and fibroblasts and macrophage cells mainly interacted through TGFB3-(TGFBR1+TGFBR2), and TGFB1-(TGFBR1+TGFBR2) (Fig. 4C). In the TGF-β signaling pathway, the intensity of communication mediated by TGFB3 with its receptors TGFBR1 and TGFBR2 was higher, mainly between fibroblast cells and macrophage cells as well as NK/T cells (Fig. 4D). TGFB1-mediated communication with TGFBR1 and TGFBR2 on the other hand functioned mainly between fibroblast cells and macrophage cells (Fig. 4E). In contrast, communication mediated by TGFB3 with ACVR1 and TGFBR1 was weaker and occurred mainly between fibroblast cells and macrophage cells and NK/T cells (Fig. 4F). Overall, fibroblast cells showed stronger cellular communication in the TGF-β signaling pathway with macrophage cells and NK/T cells.

Figure 3 The TGF BETA signaling pathway inhibits immune activity, promotes escape, and regulates metabolism.

(A) Correlation of TGF BETA SIGNALING with immunosuppressive signaling pathway activity. (B) Correlation of TGF BETA SIGNALING with proliferative signaling activity. (C) Correlation of TGF BETA SIGNALING with metabolic signaling pathways.

Figure 4 Stronger TGF BETA signaling communication between fibroblasts and macrophage cells and NK/T cells.

(A) Number of interacting ligand–receptor pairs between fibroblasts and seven cell types. (B) TGF BETA SIGNALING-based cellular communication in fibroblasts and seven cell types. (C) TGF BETA SIGNALING receptor-mediated cellular communication. (D) TGFB3-(TGFBR1+TGFBR2) receptor-mediated cellular communication. (E) TGFB1-(TGFBR1+TGFBR2) receptor-mediated cellular communication. (F) TGFB3-(ACVR1+TGFBR1) receptor-mediated cellular communication.

WGCNA identification of TGF-β signaling pathway related genes module

Further, TGF-β signaling pathway-correlated genes were identified in CC by WGCNA method in TCGA-CESC dataset. First, in TCGA-CESC, we calculated the TGF-β signaling pathway score for all samples. The samples were divided into high- and low- scoring groups by the median value, and the K-M curves of the two groups of patients could be visualized to observe that the low-scoring patients showed a significant survival advantage (Fig. 5A). The co-expression network was then identified by the WGCNA method. When the soft threshold β = 6, the network was scale-free (Fig. 5B). Gene module identification was performed by hierarchical clustering, and 17 co-expression modules were generated after merging the modules, and it is worth noting that grey modules were genes that could not be clustered into other modules (Fig. 5C). Finally, by analyzing the correlation between the first principal component eigenvectors of the 17 gene modules and the TGF-β signaling pathway scores, the brown module showed a significant positive relationship with the TGF-β signaling pathway scores (r = 0.51, p = 8.39e−21) (Fig. 5D). Overall, we identified brown modules in the TCGA-CESC dataset, which was remarkably positively related to the TGF-β signaling pathway.

Figure 5 WGCNA identification of gene modules related to TGF-β signaling pathway.

(A) K-M curves of patients in the high and low TGF-β signaling pathway score subgroups in the TCGA-CESC dataset. (B) Network connectivity under different soft threshold parameters. (C) Gene dendrogram based on 1-TOM clustering. (D) Heatmap of pearson correlation between first principal component eigenvectors of gene modules and TGF BETA signaling pathway score.

Prognostic diagnostic model for CC

Three prognostically relevant genes in CC were identified by univariate COX, LASSO COX, and multivariate COX analyses in the TCGA-CESC training set (Figs. 6A–6B). We developed a gene assessment model for evaluating the prognosis of CC with risk score=0.539*ITGA5 + 0.403*SHF − 0.311*SNRPN. The median value of the risk score was calculated to group patients in the TCGA training set, the TCGA validation set, and the entire TCGA-CESC cohort into the high risk score group and low risk score group. In all three cohorts, patients in the low risk score group showed a better survival advantage as seen in the K-M curves of patients in both groups (Figs. 6C–6E). In all three cohorts, ROC curves predicting patients’ survival at 1, 2, 3, 4, and 5 years based on risk score showed favorable AUC values (Figs. 6F–6H). Collating the survival of patients in the TCGA-CESC cohort, the number of patient deaths in the high risk score group was also higher than that in the low risk score group (Fig. 6I). To further validate the model’s robustness, we validated the external validation set GSE44001 dataset using the same method. The low risk score group also showed a significant survival trend (Fig. 6J). The risk score model in the GSE44001 dataset also showed good AUC values (Fig. 6K).

Figure 6 Prognostic diagnostic models for CC.

(A) Trajectory plots of penalty parameter Lambda selection and independent variable variation with Lambda in LASSO COX analysis. (B) Forest plot of the results of multivariate COX analysis. (C) K-M curves of patients grouped by high and low risk score subgroups in the TCGA training set. (D) K-M curves of patients in the two subgroups in the TCGA validation set. (E) K-M curves of patients in the two subgroups in the TCGA-CESC dataset. (F) ROC curves in the TCGA training set for risk score-predicted 1-, 2-, 3-, 4-, and 5-year survival of patients. (G) ROC curves in the TCGA validation set for risk score-predicted patient survival at 1, 2, 3, 4, and 5 years. (H) ROC curves in the TCGA-CESC dataset for risk score prediction of patient survival at 1, 2, 3, 4, and 5 years. (I) Survival status of patients in the two subgroups in the TCGA-CESC cohort. (J) K-M curves of patients in the two subgroups in the GSE44001 dataset. (K) ROC curves for risk score-predicted 1-, 2-, 3-, 4-, and 5-year survival of patients in the GSE44001 dataset.

Immune infiltration characteristics in high- and low-risk score groups

The characteristics of immune infiltration in each group were analyzed by three immune infiltration analysis methods to compare the differences in the immune microenvironment between different risk score subgroups. According to the ESTIMATE results, patients in the low risk score showed a higher ImmuneScore (Fig. 7A). The distribution of the six immune cell scores in different subgroups in the TCGA-CESC dataset was calculated based on the TIMER database. The results demonstrated that the scores of CD4_Tcells and B_cells were significantly higher in the low risk score group, suggesting that the high activity of these immune cell types in the low-risk group may play a key role in activating the immune system, killing tumor cells, and inhibiting cancer progression (Fig. 7B). In addition, by performing ssGSEA, and it was found that risk score and immature B cell, activated dendritic cell, macrophage, activated CD8 T cell, and activated B cell all showed a significant negative correlation trend, indicating that the level of immune infiltration decreased significantly as the risk score increased (Fig. 7C). The above findings further support that an active immune environment in the low-risk group may be important for tumor suppression.

Figure 7 Characterization of immune infiltration in high and low risk score groups.

(A) ImmuneScore, StromalScore, ESTIMATEScore of patients in high and low risk score subgroups. (B) Infiltration scores of six immune cells assessed based on the TIMER method in the two subgroups. (C) Heatmap of correlation between scores of 28 types of immune cell and risk score. *p < 0.05, **p < 0.01, ***p < 0.001, ****p < 0.0001, and ns stands for no significant difference.

Cellular validation based on CC cells

The involvement of ITGA5, SHF and SNRPN in CC was additionally explored via a series of cellular validation assay. The mRNA levels of the three genes were firstly calculated, and it was seen that the expressions of ITGA5 and SNRPN were higher yet that of SHF was lower in CC cells Hela as compared to those in Ect1/E6E7 cells (Fig. 8A, p < 0.05). In addition, we found, based on CCK-8 and EdU assays, that cell viability and the number of EdU-positive cells were significantly lower in the si-ITGA5 group compared to the control group (Figs. 8B–8C, p < 0.01). Next, the effects of ITGA5 silencing on the migration and invasion of CC cells were investigated. The relevant results suggest that silencing ITGA5 leads to attenuation of the invasive and migratory capacity of Hela cells (Figs. 8D–8E, p < 0.01).

Figure 8 Cellular validation results.

(A) Quantified mRNA levels of ITGA5, SHF and SNRPN in CC cells Hela and cervical epithelial cells Ect1/E6E7. (B) Determination on the viability of transfected CC cells Hela based on the CCK-8 assay. (C) The effect of ITGA5 silencing on the proliferative capacity of Hela cells was examined by EdU assay. (D) The effect of ITGA5 silencing on the invasive ability of Hela cells was assessed by transwell assay. (E) The effect of ITGA5 silencing on the migratory capacity of Hela cells was assessed by a closure healing assay. *p < 0.05, **p < 0.01, and ***p < 0.001.

Discussion

Fibroblasts in the TME are one of the significant factors promoting tumor progression, and they remodel the extracellular matrix, promote angiogenesis and suppress the immune system by secreting various cytokines and growth factors (Biffi & Tuveson, 2021). The TGF-β signaling pathway can promote tumor metastasis by mediating epithelial-mesenchymal transition (Hao, Baker & Ten Dijke, 2019). TGF-β protein secreted by tumor cells was found to activate (ERK)1/2 signaling in cancer-associated fibroblasts to produce more CLCF1 leading to tumor progression (Song et al., 2021). Investigating the mechanism of TGF-β signaling pathway in fibroblasts may help to improve the prognosis and treatment of CC. We found that genes specifically highly expressed in fibroblasts (COL1A2, COL1A1, and DCN) exhibited a significant up-regulation in TME of CC. Cancer-associated fibroblasts are important contributors to the EMT featured in the tumor microenvironment, and the EMT phenomenon and the level of COL1A1 and COL1A2 were closely correlated (Szabo et al., 2023). Down-regulation of DCN expression resulted in the activation of the IL-6/STAT3/AUF1 signaling pathway, and endogenous DCN could inhibit breast stromal fibroblasts’ pro-metastatic and pro-carcinogenic effects (Aljagthmi et al., 2024). The significant up-regulation of COL1A2, COL1A1 and DCN in the TME of CC illustrates the phenomenon of fibroblasts’ activation in the tumor microenvironment. The high expressions of these genes are closely correlated with the formation of extracellular matrix, which in turn may lead to the phenomenon of tumor invasion and metastasis.

We found significant activation of the TGF-β signaling pathway in fibroblasts of CC samples, which was strongly associated with cancer progression and poor prognosis. The TGF-β signaling pathway has shown a dual role in the tumor microenvironment, where it both inhibits early tumorigenesis and promotes tumor invasion and metastasis during the stage of tumor progression (Batlle & Massague, 2019). Fibroblasts are highly heterogeneous stromal cells, and in pancreatic ductal adenocarcinoma, TGF-β is a major driver of cancer-associated fibroblasts (Mucciolo et al., 2024; Wu et al., 2021). In the advancement of CC, it was found that TGF-β signaling interacting with localized regions of cancer-associated fibroblasts could promote the invasive phenomenon of CC cells (Nagura et al., 2015). In our results, there was a significant TGF-β signaling exchange between fibroblasts and NK cells and macrophages. TGF-β was found to inhibit NK cell activation and function by inhibiting the mTOR pathway, specifically, TGF-β signaling or mTOR depletion prevented NK cell development (Viel et al., 2016). In the presence of inhibitors of TGF-β signaling, the tumor-killing effect of NK cells was improved (Shaim et al., 2021). TGF-β proteins can also modulate the activity of tumor-associated macrophages stimulating tumor proliferation and even leading to tumor immune escape (Gratchev, 2017). TGF-β signaling in the TME also promotes macrophage M2 polarization, and TGF-beta activates fibroblasts to form CAFs that secrete a higher level of CXCL12, which binds to the cognate receptor CXCR4 on M2 macrophages, leading to tumor cell growth (Wu et al., 2022). This further supports that the TGF-β signaling pathway in the CC tumor microenvironment may accelerate tumor progression and lead to poor prognosis through interactions with fibroblasts, NK cells, and macrophages. The current study constructed a prognostic gene model of CC with three prognostic genes (ITGA5, SHF, and SNRPN). It was indicated that ITGA5 could promote CC angiogenesis and ITGA5 might be a poor prognostic biomarker for CC patients (Xu et al., 2023). Down-regulation of ITGA5 expression reduced the proliferation and invasion ability of CC cells (Yao et al., 2023). SHF is a tumor suppressor of glioblastoma and inhibits glioblastoma progression by negatively regulating STAT3 dimerization (Wang et al., 2022). These genes are involved in tumor progression and prognosis, and they could be used as potential biomarkers for evaluating the prognosis of CC patients. The risk score showed good prognostic accuracy in the TCGA-CESC cohort, and the GSE44001 dataset, which further demonstrated the robustness of the model. In addition, the risk score performed relatively well in evaluating 1-, 2-, 3-, 4-, and 5-year survival, suggesting that the model has high predictive accuracy at different time points.

However, there are some limitations to our study. First, this study is based only on multi-omics data obtained from public databases, and the analysis is mainly based on the RNA level of inquiry, for this reason, it is necessary to further combine the data with other levels of DNA methylation, mutation profiles, etc., in order to fully understand the mechanism of gene regulation. In addition, the expansion of data sources, as well as the inclusion of multicenter and diverse samples of more patients to enable in-depth validation of the applicability of the study results. Finally, this study was based on an in vitro cellular model for validation. In the future, we will carry out a mouse xenograft tumor model, as well as collect more clinical samples to validate the expression of key genes and their correlation with clinical prognosis in combination with immunohistochemistry.

Conclusion

Overall, we offered new perspectives for understanding molecular mechanism of CC and provides a theoretical foundation for developing therapeutic strategies and personalized prognostic assessment tools based on the TGF-β signaling pathway. These findings not only help to further investigate the pathobiological features of CC, but also provide new chances for future clinical applications.

Supplemental Information

Supplemental Information 1 MIQE checklist

Abbreviations

WHO World Health Organization

CC Cervical cancer

TME tumor microenvironment

CAFs cancer-associated fibroblasts

WGCNA weighted gene co-expression network analysis

EMT epithelial-mesenchymal transition

CESC Cervical squamous cell carcinoma and endocervical adenocarcinoma

TCGA the cancer Genome Atlas

GEO Gene Expression Omnibus

MsigDB Molecular Signatures Database

PCA principal component analysis

UMAP Uniform Manifold Approximation and Projection

K-M Kaplan–Meier

ROC Receiver Operating Characteristic Curve

AUC area under the curve

ssGSEA single sample gene set enrichment analysis

CCK-8 cell counting kit-8

Additional Information and Declarations

Competing Interests

Author Contributions

Data Availability

The authors declare there are no competing interests.

Haina Qu conceived and designed the experiments, analyzed the data, authored or reviewed drafts of the article, and approved the final draft.

Jing Zhao conceived and designed the experiments, authored or reviewed drafts of the article, and approved the final draft.

Xia Zuo performed the experiments, analyzed the data, prepared figures and/or tables, and approved the final draft.

Hongyue He conceived and designed the experiments, prepared figures and/or tables, and approved the final draft.

Xiaohan Wang conceived and designed the experiments, analyzed the data, prepared figures and/or tables, and approved the final draft.

Huiyan Li performed the experiments, authored or reviewed drafts of the article, and approved the final draft.

Kun Zhang performed the experiments, analyzed the data, authored or reviewed drafts of the article, and approved the final draft.

The following information was supplied regarding data availability:

The datasets generated during and/or analyzed during the current study are available at GSE: GSE44001 and GSE208653.

The raw data is available at GitHub and Xenodo:

– https://github.com/21kunzhang/raw-data.git.

– 21kunzhang. (2024). 21kunzhang/raw-dat(A) Updated raw data (v.1.1.1). Zenodo. https://doi.org/10.5281/zenodo.14532964.

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
