# Peer review of "TGF-β-mediated activation of fibroblasts in cervical cancer: implications for tumor microenvironment and prognosis"

_PeerJ, doi:10.7717/peerj.19072_

## Round 0.1 · original submission · Major Revisions

The rationale for using ect1/e6e7 cells needs explanation, and experimental conditions should be clearly described.
Several figures require improvement in quality and clarity, particularly Figures 2b, 2f, 2k, 2j, 3c, and 6c. Statistical significance indicators are missing in multiple figures. The methodology for dividing TCGA training and validation sets needs clear description.
Abbreviations should be properly defined at first use, and terminology (e.g., TGF-β/TGF BETA) should be standardized throughout. The abstract's background section needs condensing, and some introduction content appears irrelevant to the study's focus.

Reviewer 1 ·

Basic reporting

no comment

Experimental design

no comment

Validity of the findings

no comment

Additional comments

The authors attempt to verify the roles of TGA5, SHF, and SNRPN in CC, but the experimental design is incomplete. The authors need to supplement the experiments with the following:
1) Supplement the CCK8 assay with more time points, or add Edu assay results. Since the CCK8 assay has large errors and is unstable, more experimental results are needed to prove the correctness of the conclusions.
2) It is recommended to add wb and qPCR experiments to further validate the function of the genes and pathways.

Other comments:

1. The background in the abstract is too long and needs to be streamlined.
2. "The main reasons for its high incidence are related to the high prevalence of HPV (human papillomavirus) infection, low screening coverage, and limited resources for treatment" is not recommended for introduction in the introduction, as it seems irrelevant to this study.
3. In Fig3c, are the x-axes of the first row in the same order as the second row? I do not recommend this way of presentation; I suggest labeling the x-axis of the first row as well for better standardization.
4. It appears that Fig3c lacks the expression results for COL1A2 and DCN?
5. Fig2b lacks significance symbols, making it impossible to determine if there are significant differences in the results.
6. Fig2f is too blurry; please provide a high-resolution image that allows the identification of gene names.
7. Fig2k and j lack significance symbols, making it impossible to conclude which group is highly expressed.
8. "REGULATIONOFCD4POSITIVEALPHABETATCELLDIFFERENTIATION," please change to lowercase and remove the underline.
9. For simplicity and clarity, the authors can choose to describe only the important pathways in the main text for Fig3.
10. The significance symbols in Fig7a are not fully displayed.
11. The reason for using ect1/e6e7 cells in Fig6a needs to be explained in the methodology.
12. In FigB, all experimental group conditions and the meanings of group abbreviations need to be explained in the methodology.
13. Fig6c is too blurry to discern the specific contours of the cells.

Reviewer 2 ·

Basic reporting

The current study, through multi-omics analysis integrated with cellular validation assays, investigates the role of TGF-ꞵ signaling pathway in cervical cancer (CC), explores its interaction with other immune cells, and constructs a prognostic RiskScore model. The RiskScore model could effectively assess the prognosis of CC patients. Moreover, ITGA5 knockdown could suppress the viability, migration and invasion of CC cells. This study could provide new perspectives for further research on the pathological mechanisms and clinical management of CC. On the whole, the experimental design of this study is logical, and the research content has certain scientific significance. But the manuscript writing quality is poor.
1. There are several abbreviation definition problems in the manuscript. For instance, the abbreviation of cervical cancer has been defined in front Line51, which can be directly expressed by CC in Line54, 60, 73, 80, etc. The full name of “TME” should be defined when it first appears (Line67), not in Line81. The full name of “CAFs” (Line78), “ECM” (Line86), “OS” (Line117), “UMAP” (Line127), “WGCNA” (Line150) and so on should be supplemented. Please follow the principle of first appearance to check and modify the full text.
2. The handwriting of “TGF-ꞵ”, “TGF BETA”, “TGF-beta” is not uniform in the full text, in order not to confuse readers, please specify a form for modification. In addition, the spelling of “Person method” (Line137) is incorrect, which should be “Pearson”. The gene name of ITGA5, SHF, SNRPN, etc should be written in italics.
3. The “with cervical cancer” in the sentence of “by analyzing the multi-omics data of CC patients with cervical cancer” (Line102-103), the “Phenotypic data.” (Line111) seems redundant.
4. In the Materials and Methods 2.3 section, the title is ‘Correlation analysis of TGF_BETA_SIGNALING activity with immune pathways, proliferation and metabolism’. While, in Line135-136, the description is ‘we extracted the inflammatory, proliferative and metabolic signaling pathways from MsigDB’. The outcomes of this study appear to be not involved in the inflammatory pathways.
5. Method description of 2.7 section is not clear, what is “these genes” of the sentence ‘The prognostic relevance of these genes was determined by univariate COX regression analysis,’ indicates? In Line163-165, “the receiver operating characteristic curve (ROC) was plotted by Kaplan-Meier (K-M) survival analysis based on the area under the curve (AUC) estimation model to predict the performance of prognosis.” ROC analysis and K-M survival analysis are two different curves, which should not be mix up. It is recommended to separate description. Additionally, the formula of RiskScore is required to add into the manuscript.
6. As for immune cell infiltration analysis of this study, the used methods are differed in the previous and later articles. Line170-171 is ‘Ten immune cell scores were calculated by the MCPcounter package (29).’, while Line338 is ‘calculated based on the TIMER database’ and Figure 7B legend is ‘6 immune cells assessed based on the TIMER method’, please modify it.
7. For the results of Fig.6C-H, how are the TCGA training set and TCGA validation set divided, and how many samples do they contain? Please clarify in the Methods section.
8. In the results of 3.8 section, the Figure number is miswritten, please correct them. Moreover, the significant difference asterisks “*” need to be annotated in the legends of Figure 6B, Figure 7, and Figure 8.
9. The conclusion that TGF-ꞵ signaling pathway promotes immune escape (Section3.3 and Line390-392) is based on what results can be obtained? This current study does not seem to have any results or experiments related to immune escape.
10. It is suggested to add a paragraph at the end of the discussion section about the limitations of this study and the prospects for the future.

Experimental design

no comment

Validity of the findings

no comment

---

## Round 0.2 · accepted · Accept

Authors have addressed all comments and this paper has been significantly improved. I think this paper can be accepted for publication.

Reviewer 1 ·

Basic reporting

The author has added sufficient experiments and addressed my comments, and I believe they can be published in their current form.

Experimental design

no comment

Validity of the findings

no comment

Reviewer 2 ·

Basic reporting

The author systematically studied the role of the TGF-β signaling pathway through multi-omics analyses, explored its interactions with other immune cells, and constructed a prognostic RiskScore model. The RiskScore model can effectively evaluate the prognosis of CC patients. In addition, ITGA5 knockout can inhibit the survival, migration, and invasion of CC cells. This study can provide a new perspective for further research on the pathological mechanisms and clinical management of CC. Overall, the experimental design of this study is logical and the research content has certain scientific significance. The manuscript can be published in its current form.

Experimental design

no comment

Validity of the findings

no comment